

# Extended characterization of petroleum aromatics using off-line LC-GC-MS

Khoa Huynh, Annette E. Jensen and Jonas Sundberg

Danish Hydrocarbon Research and Technology Centre, Technical University of Denmark, Kgs. Lyngby, Denmark

## ABSTRACT

Characterization of crude oil remains a challenge for analytical chemists. With the development of multi-dimensional chromatography and high-resolution mass spectrometry, an impressive number of compounds can be identified in a single sample. However, the large diversity in structure and abundance makes it difficult to obtain full compound coverage. Sample preparation methods such as solid-phase extraction and SARA-type separations are used to fractionate oil into compound classes. However, the molecular diversity within each fraction is still highly complex. Thus, in the routine analysis, only a small part of the chemical space is typically characterized. Obtaining a more detailed composition of crude oil is important for production, processing and environmental aspects. We have developed a high-resolution fractionation method for isolation and preconcentration of trace aromatics, including oxygenated and nitrogen-containing species. The method is based on semi-preparative liquid chromatography. This yields high selectivity and efficiency with separation based on aromaticity, ring size and connectivity. By the separation of the more abundant aromatics, *i.e.*, monoaromatics and naphthalenes, trace species were isolated and enriched. This enabled the identification of features not detectable by routine methods. We demonstrate the applicability by fractionation and subsequent GC-MS analysis of 14 crude oils sourced from the North Sea. The number of tentatively identified compounds increased by approximately 60 to 150% compared to solid-phase extraction and GC × GC-MS. Furthermore, the method was used to successfully identify an extended set of heteroatom-containing aromatics (*e.g.*, amines, ketones). The method is not intended to replace traditional sample preparation techniques or multi-dimensional chromatography but acts as a complementary tool. An in-depth comparison to routine characterization techniques is presented concerning advantages and disadvantages.

Corresponding author
Jonas Sundberg, jonsun@dtu.dk

# INTRODUCTION

The use of petroleum as a feedstock for energy production is declining. However, certain critical functions cannot safely be replaced by renewable energy (*International Energy Agency, IEA)(2021*). Secondly, petroleum is a fundamental feedstock for the production of a large number of chemical starting materials (*Aftalion, 2001*; *Yadav, Yadav & Patankar, 2020*). Therefore, reducing the environmental impact of oil production is an important goal. This requires a better understanding of petroleum on the molecular level. Crude

oil is a complex mixture of saturated and aromatic hydrocarbons with a smaller fraction of heteroatom-containing compounds, *i.e.*, the resins and asphaltenes. The molecular distribution typically ranges from 16 to 1,000 amu (*Marshall & Rodgers, 2008*). The number of unique compounds is extensive and more than 240,000 molecular species have been resolved in a single sample (*Krajewski, Rodgers & Marshall, 2017*; *Palacio Lozano et al., 2019*). Due to this complexity, a large portion of the petroleum chemical space is structurally unknown.

We have previously looked at the resins fraction (*i.e.*, polar heteroatom-containing species) of North Sea oils (*Sundberg & Feilberg, 2020*). Herein, we extend our work with a focus on aromatics. Within this fraction, the dominant species (in terms of abundance) are monoaromatic followed by a smaller amount of polycyclic aromatic hydrocarbons (PAHs) (*Requejo et al., 1996*; *Wei et al., 2018*). The PAHs class is dominated by smaller (2 to 3 rings) PAHs, with larger species (*e.g.*, chrysene, coronene) being present at trace levels. It also contains small amounts of heteroatomic-containing ring structures (*Mössner & Wise, 1999*; *Zhang et al., 2018*; *Carvalho Dias et al., 2020*). Due to their toxicity, PAHs have been extensively studied (*Lawal, 2017*). A large focus has been on the 16 priority pollutants PAHs defined by the US Environmental Protection Agency (*Keith, 2015*). However, this list is not representative of crude oils which contain a more structurally diverse PAH set (*Andersson & Achten, 2015*; *Stout et al., 2015*, p. 16). Low molecular weight PAHs are susceptible to weathering, primarily by volatilization, whereas high molecular weight aromatics are more resilient (*John, Han & Clement, 2016*). Therefore, these are useful targets for oil-oil and oil-source correlation and spill identification and environmental monitoring (*Pampanin & Sydnes, 2017*; *Poulsen et al., 2018*).

Comprehensive identification of the aromatics is challenging due to the large concentration variance. Traditionally, petroleum analysis is based on pre-fractionation using silica chromatography or solid-phase extraction (SPE) cartridges followed by GC-MS$^n$ (*Wang, Fingas & Li, 1994*; *Alzaga et al., 2004*; *Pillai et al., 2005*; *Gilgenast et al., 2011*). SPE is a low-efficiency separation technique, based on chemical selectivity. This allows crude isolation of the aromatics fraction, but not separation of the compounds within it (Fig. 1). Thus, an aromatic fraction obtained by SPE contains both the benzenes, naphthalenes and larger rings. Here, a typical crude oil will have a high abundance of monoaromatics, with diminishing concentrations with increasing ring size. The appropriate GC on-column concentration of the naphthalenes typically results in the larger ring systems being below the limit of detection (LOD). By increasing concentration to push trace aromatics above the LOD, both the column and detector will be saturated by the more abundant compounds. This leads to high background levels which affect quantitation and may obscure mass spectra complicating structural identification of unknowns (*Zhao et al., 2014*; *Wilton, Wise & Robbat, 2017*). Furthermore, the poor resolution of SPE often leads to an overlap between the saturated and aromatic hydrocarbons, which interferes with subsequent analysis. Thus, although SPE is efficient for routine applications, a large portion of the sample remains undetected. Comprehensive multi-dimensional chromatography (GC × GC) is often used as an alternative to simplify or remove the need for sample pre-fractionation (*Nizio, McGinitie & Harynuk, 2012*; *Jennerwein et al., 2014*; *Stilo et al., 2021*). However, it does not
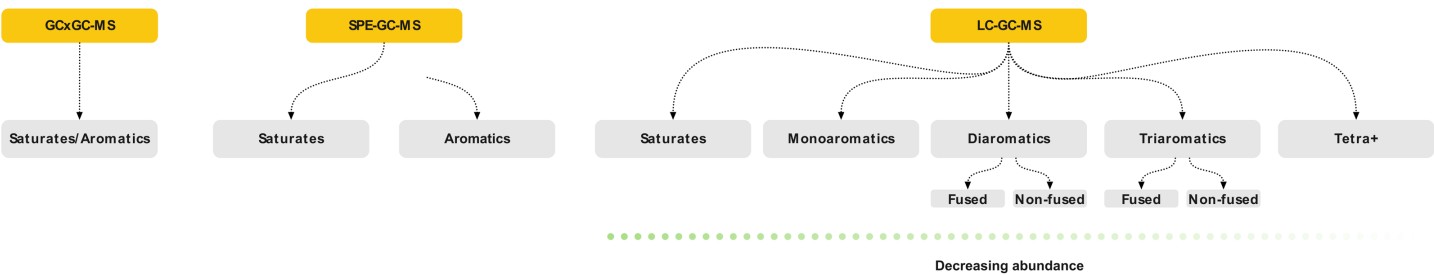

**Figure 1** Schematic of the three sample preparation and analytical strategies discussed in the paper showing the extent of fractionation possible. Grey boxes indicate the individual samples that are characterized by mass spectrometry.

solve the issue with variable abundance and column/detector overload. Thus, a complete qualitative and explorative oil analysis requires a more selective sample-prefractionation method.

Herein, we present a high-performance liquid-chromatography (HPLC) method for the automated high-resolution fractionation of crude oil using commercially available columns. The method can resolve aromatics based on ring size and connectivity, *i.e.*, fused and non-fused rings (*e.g.*, *naphthalene versus biphenyl*). The fractions may be diluted or concentrated, depending on the target, for subsequent analysis and can thus be used to concentrate trace species. The method is easily modified to selectively collect only fractions of interest, and the aromatics may be collected either as one or several fractions. We demonstrate the method's applicability by fractionation of fourteen crude oils with subsequent GC-MS analysis. The method is compared to data obtained using SPE and GC × GC-MS. We demonstrate how it's especially suitable for the analysis and identification of trace aromatics by the successful tentative identification of several compounds not observed using comparable methods.

## MATERIALS & METHODS

### Chemicals and reagents

Chloroform, dichloromethane, *n*-hexane, deuterated standards and model compounds (ethylbenzene, naphthalene, biphenyl, phenanthrene, 1-benzylnaphthalene and chrysene) were purchased from Sigma Aldrich and used as received.

### Samples

Fourteen crude oils sampled from producing fields in the Danish region of the North Sea were obtained from Mærsk Oil (now Total E&P). The samples were received in metal containers (jerrycans) and transferred to glass bottles upon arrival. The samples were stored at room temperature protected from light.

### Sample preparation
#### *Solid-phase extraction*

Crude oil (10 µL) was combined with 100 µL of a solution containing alkane internal standards (decane-D22, hexadecane-D34 and eicosane-D42, 400 µg/mL in *n*-hexane), 50

μL of PAH internal standards (naphthalene-D8, phenanthrene-D10, acenaphthene-D10, chrysene-D12 and perylene-D12, 30 μg/mL in $n$-hexane) and further diluted with $n$-hexane (840 μL). A solid-phase extraction column (Phenomenex EPH Strata, 200 μm, 70 Å, 500 mg / three mL) was cleaned and conditioned by $CH_2Cl_2$ (3 × 1 mL) followed by $n$-hexane (3 × 1 mL). 100 μL of oil solution was carefully applied to the column and was allowed to settle for 5 min. Saturated hydrocarbons were eluted into one fraction with three portions of $n$-hexane (3 × 600 μL). Aromatic hydrocarbons were eluted using dichloromethane (1× 1,800 μL). The solvent level of each fraction was reduced to 500 μL under a gentle stream of nitrogen without applied heating to avoid losses of volatile components.

### Liquid chromatography fractionation

Fractionation of crude oil was carried out on a Dionex UltiMate 3000 HPLC equipped with a DAD-3000 diode array, a RefractoMax RI-521 refractive index (RI) detector and an AFC-3000 fraction collector. The system was fitted with one six-port/two-way and one ten-port/two-way port to enable selective backflush of the primary column. A Thermo Scientific Hypersil Gold NH$_2$ (4.6 mm i.d., 3 μm, 150 mm) and a Hypersil Silica (4.6 mm i.d., 3 μm, 150 mm) were connected in series. The sample manager was kept at 20 °C and the column oven at 30 °C. The injection volume was 50 μL.

Samples were diluted at 1:2000 in $n$-hexane and stored at −20 °C for 24 h to precipitate asphaltenes. The samples were centrifuged and an aliquot of the mother liquor was carefully transferred to an autosampler vial for analysis. Separation of saturates and aromatics was achieved via isocratic $n$-hexane elution during which 30 s wide fractions were collected. After elution of aromatics, the primary column was rinsed using a backflush gradient from $n$-hexane to 1:1 2-propanol:chloroform. The collected fractions were diluted (saturates, mono- and di-aromatics) or concentrated (tri-aromatics and larger) for analysis on GC-MS. For enrichment experiments, consecutive fractionations (typically 3 to 6) were performed with pooling of the eluents followed by solvent reduction under a gentle stream of N$_2$ at 30 °C.

## Analytical methods
### GC-MS

GC-MS data were recorded using an Agilent 5977B GC-MSD as follows; 250 °C inlet, 320 °C transfer line, splitless injection (1 μL), Agilent DB-5MS (30 m, 0.25 mm i.d., 0.25 μm). The oven temperature gradient was programmed as follows; 50 (1 min. hold-time) - 320 °C (8 min hold-time, 10 °C/min.), helium carrier gas at 1.5 mL/min. in constant flow mode.

GC × GC-MS data were recorded using an Agilent 7200B GC-QTOF equipped with a Zoex ZX-2 thermal modulator (Zoex Corporation, Houston, TX, USA) as follows; 250 °C inlet, 320 °C transfer line, splitless injection (1 μL), Agilent DB-5MS UI (1D, 30 m, 0.25 mm i.d., 0.25 μm df) and a Restek Rxi-17Sil MS (2D, 2 m, 0.18 mm i.d., 0.18 μm df) capillary columns connected using a SilTite μ-union. The oven was temperature programmed as follows; 50 (1 min hold-time) - 320 °C (3 °C/min.), helium carrier gas at one mL/min. in constant flow mode. The modulation period was set to 6 s with a 400 ms hot-jet duration.

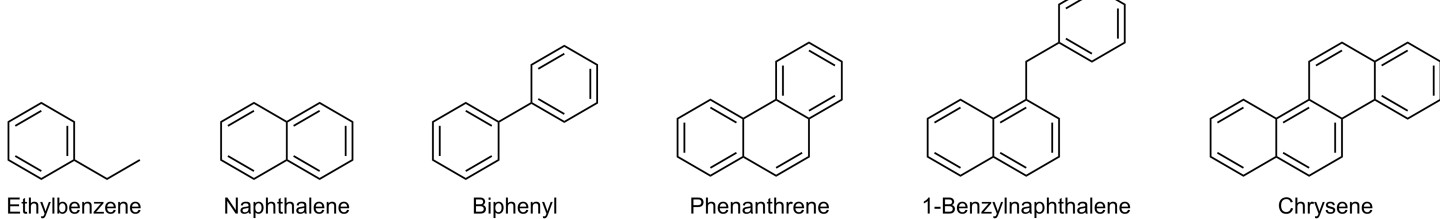

| Ethylbenzene | Naphthalene | Biphenyl | Phenanthrene | 1-Benzylnaphthalene | Chrysene |

**Figure 2** Molecular structure of the model compounds used for the HPLC method development and validation.

### Data processing

Data were screened using Masshunter Qualitative Navigator (Agilent, B.08.00). Peak detection and compound identification were performed using MassHunter Unknowns Analysis (Agilent, B.09.00) and the NIST Standard Reference Database (1A v17, Gaithersburg, MD, USA). Feature tables were exported as CSV files and imported into a Jupyter Notebook for further processing using the Python scripting language. Duplicates based on the CAS number were removed from the feature tables. All compounds containing silica and halogens were removed. The double-bond equivalent values were calculated for each compound and all features with a DBE of less than 4 were excluded. Finally, experimental and literature retention indices (RI) were compared with flagging of all compounds where the difference was larger than 50 units.

## RESULTS AND DISCUSSION

### Method development

The objective of the method was to (1) separate saturates and aromatics and (2) intra-class separation of the aromatics with enrichment capabilities. A dual-column setup using normal phase analytical LC-columns provided the required selectivity and efficiency. The primary column (*Thermo Scientific Hypersil Gold NH$_2$*) acted as a retainer for polar components, whereas a secondary pure silica-based column (*Thermo Scientific Hypersil Silica*) was required for the separation of saturated and aromatic hydrocarbons. The separation was optimized using six model compounds commonly found in crude oil (Fig. 2). An isocratic *n*-hexane elution yielded separation of saturated and mono-aromatic hydrocarbons, as well as separation of polycyclic aromatics based on ring size and connectivity (Fig. 3).

The fraction collector was programmed to collect 30 s wide fractions based on the peak widths of the model compounds. At this fraction width, we observed only a minor overlap of fractions with co-elution of the most abundant components. A reduction of the fraction width can be set if a higher peak purity is required. The cost is a slight loss of recovery. After elution of the last aromatics as observed by UV/Vis, the flow path was selectively reversed for the primary column. The column was then rinsed using a gradient from 100% *n*-hexane to 50:50 chloroform:2-propanol. This effectively removed the adsorbed resins on the amide column. The fraction collector is within the flow path during all stages of chromatography and the resins may therefore be isolated for further analysis

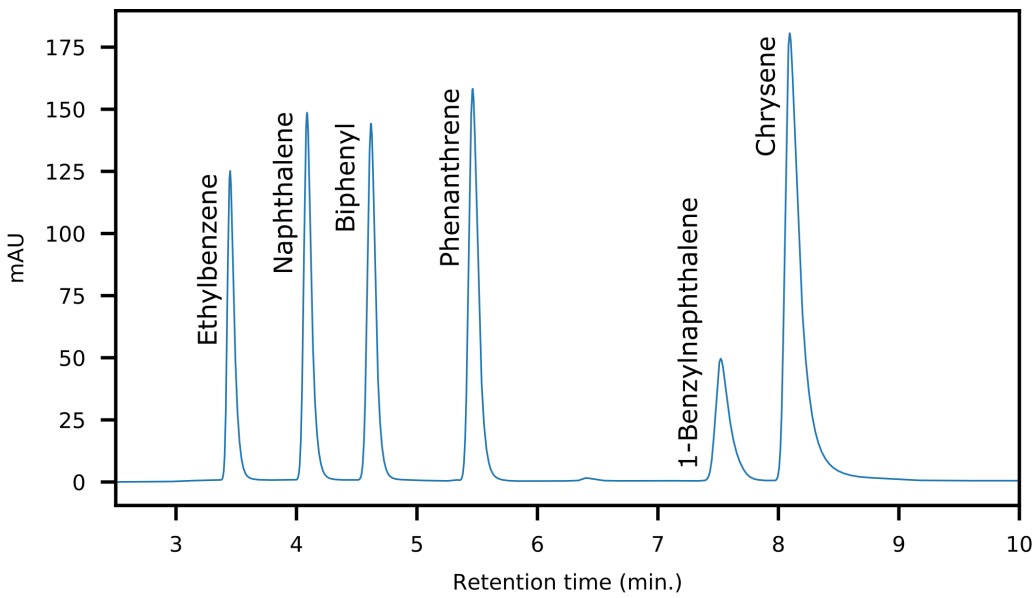

**Figure 3** HPLC-Chromatogram showing the elution order and separation of the model compounds used for method development.

(*Sundberg & Feilberg, 2020*). The final step is a re-equilibration of the whole system by a return to isocratic *n*-hexane and flushing at an increased flow rate to remove the polar solvents from the flow path. Improper re-equilibration resulted in a severe loss of retention in subsequent fractionations due to the adsorption of 2-isopropanol on the silica phase.

Recovery values were calculated using the model compounds by comparison of peak areas obtained on GC-MS from LC-fractions compared to direct analysis of the standards (Table 1). Three analytes have recovery values slightly above 100%. This is likely due to a discrepancy between programmed and real injection volume on the HPLC auto-sampler. In contrast, the recovery is less than 90% for chrysene. For this compound, we observe peak broadening due to the high capacity factor. As the fraction collection width is static during the full run, the low recovery is attributed to the peak being wider than the collection width. To evaluate the reproducibility of complex samples, a single oil was fractionated three consecutive times. Each fraction was analyzed on GC-MS and the relative standard deviation was determined from peak areas. 1,2,4-Trimethylbenzene, naphthalene and phenanthrene gave 4.6,8.1 and 6.4% respectively. The results are similar to those obtained using a model mixture. This shows that the method performs consistently in the presence of a highly complex oil matrix. Furthermore, the method successfully removes interferences and yields a high signal-to-noise ratio for the target analytes in each fraction (Fig. 4).

## Applicability in crude oil analyses

The applicability of the method was demonstrated by fractionation and analysis of 14 crude oils. The oil samples were sourced from producing fields in the Danish region of the North Sea. Crude oils from this region typically have an aromatics content of 25–30%, of which the majority are BTEX-type monoaromatics (*benzene, toluene, ethylbenzene, xylene*) with

**Table 1** Recovery calculations based on integrated area as determined by GC-MS for pure model compounds and those isolated using HPLC-fractionation.

| Model compounds | Area, STD GC-MS (%RSD, $N = 6$) | Area, STD LC-GC-MS (%RSD, $N = 6$) | Recovery (%) |
|---|---|---|---|
| Naphthalene | 3344437 (1.0%) | 3566196 (5.7%) | 106.6 |
| Biphenyl | 1525309 (1.6%) | 1627554 (2.1%) | 106.7 |
| Phenanthrene | 1145132 (3.1%) | 1162530 (4.7%) | 101.5 |
| 1-Benzylnaphthalene | 599597 (4.4%) | 592272 (3.0%) | 98.8 |
| Chrysene | 220091(5.1%) | 195914 (2.8%) | 89.0 |
| **Crude oil compounds** | **Area, SPE-GC-MS (%RSD, $N = 3$)**[*] | **Area,oil LC-GC-MS (%RSD, $N = 3$)**[*] | |
| 1,2,4-Trimethylbenzene | 29538362 (17.6%) | 12638320 (4.6%) | |
| Naphthalene | 20202722 (18.0%) | 5153076 (8.1%) | |
| Phenanthrene | 9239818 (17.4%) | 3478436 (6.4%) | |

**Notes.**
[*]The area difference is due to different dilution factors.

a continuous decrease in abundance with increasing ring size (*Sundberg & Feilberg, 2020*). The primary fraction is the saturated hydrocarbons followed by the resins (up to 5%) with only traces of asphaltenes. This is evident from the fractionation, where a typical dilution factor of 50/20 had to be applied to the saturated and monoaromatic fractions respectively (Table 2). The fractions containing larger aromatics were analyzed either undiluted or concentrated by solvent reduction.

The first fraction contains the paraffins and naphthenes and is poorly retained on the primary LC-column (Fig. 5). The second silica column is required to separate them from the monoaromatics, which elute as the second fraction (Fig. 6). The third and fourth fractions contain diaromatic species, with the latter non-fused ring systems (*e.g.*, naphthalene versus biphenyl). Fractions 5 and 6 contain the triaromatics (*e.g.*, phenanthrene versus 1-phenylnaphthalene). Here, the abundance starts to diminish and the sixth fraction had to be concentrated for subsequent GC-MS analysis. Fractions 7 and above contain larger ring systems, *e.g.*, chrysene, perylene. These fractions are less well-defined, likely because compounds eluting within this retention range are fewer in number and present in trace amounts. We also observed a slight loss of resolution, with minor overlap and cross-contamination. This is a result of two things; (1) diffusion and peak broadening during the liquid chromatography (2) collection of low abundance (undiluted/concentrated) fraction after a high abundance (diluted) fraction. If higher purity peaks are required the fraction collection width can be reduced. Attempts to concentrate fractions 9 and later were not successful and the gas chromatograms were dominated by background contaminants likely originating from the solvents, HPLC-tubing and glassware (*e.g.*, siloxanes, surfactants).

## Performance comparison with SPE-GC-MS and GC × GC-MS

Solid-phase extraction of crude oil into its saturated and aromatic fraction is a well-established sample preparation method. The physical properties of SPE adsorbents (large particle size, low mass loadings) result in limited separation power (*Berrueta, Gallo & Vicente, 1995*; *Buszewski & Szultka, 2012*). Thus, the technique is mainly applicable for the

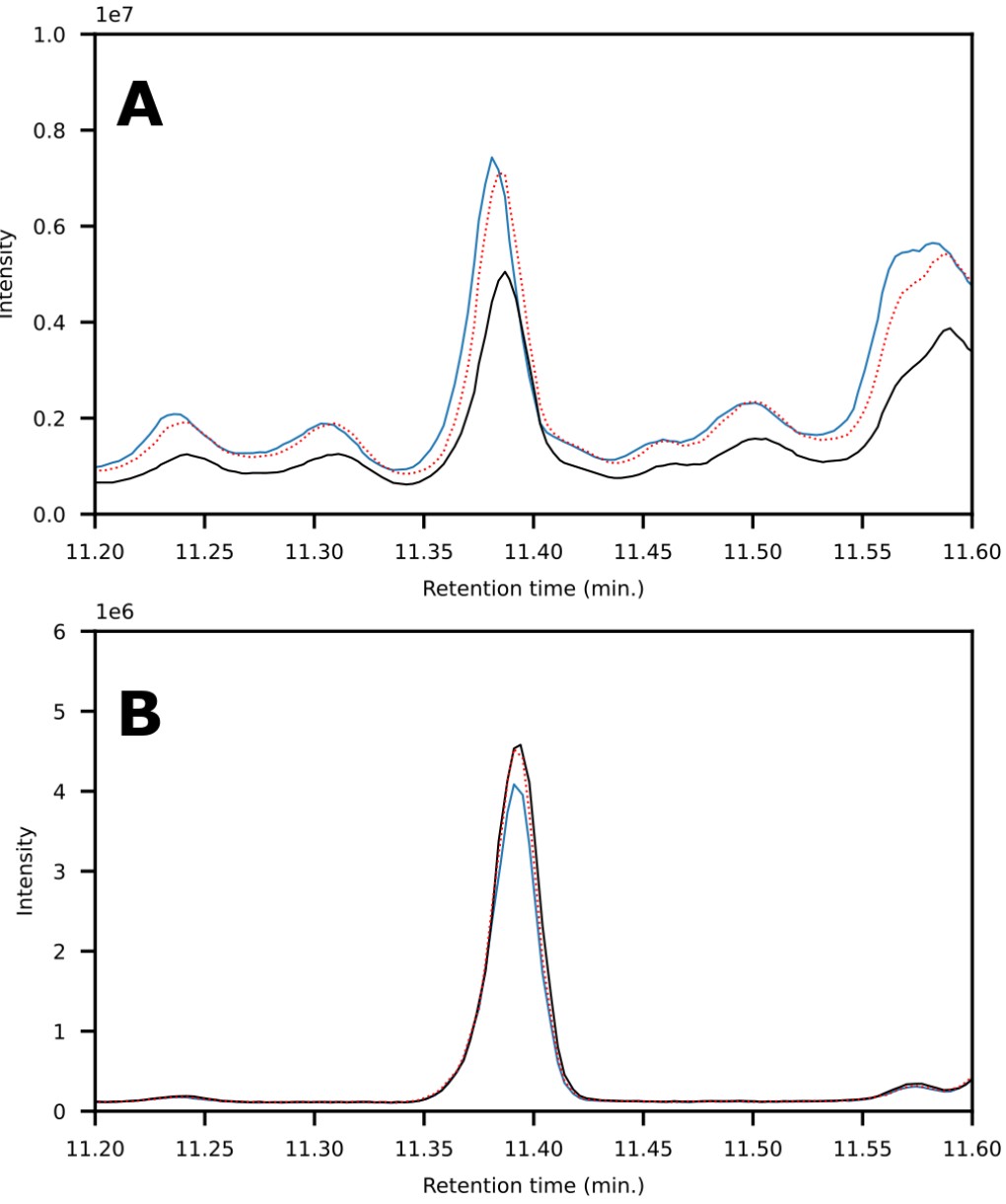

**Figure 4** **Chromatogram showing the peak corresponding to phenanthrene obtained via SPE-extractions (A) as compared to LC-fractionation (B).** The sample was fractionated using each method in triplicate to determine reproducibility. Signal intensity variance between (A) and (B) is due to different dilution factors (on-column).

crude separation of different compound classes. It does not provide sufficient resolution to separate closely related compounds within subfractions. To compare our method to SPE we fractionated each oil using a Phenomenex Strata EPH (200 μm, 70 Å, 500 mg/three mL). The cartridge contains a proprietary phase specifically developed to separate hydrocarbon fractions (*Countryman, Kelly & Garriques, 2005*).

**Table 2  Summary description of the major constituents of each fraction.**

| Fraction | Main composition | Comment |
|---|---|---|
| 1 | Saturated hydrocarbons | High concentration; diluted for GC-MS. |
| 2 | Mono-aromatics | High concentration; diluted for GC-MS. |
| 3 | Di-aromatics | Medium concentration; diluted for GC-MS. |
| 4 | Non-fused di-aromatics | Medium concentration; diluted for GC-MS. |
| 5 | Tri-aromatics | Low concentration; undiluted for GC-MS. |
| 6 | Non-fused tri-aromatics | Trace concentrations; concentrated for GC-MS. |
| 7 + 8 | Misc. tetra-aromatics | Trace concentrations; concentrated for GC-MS. |

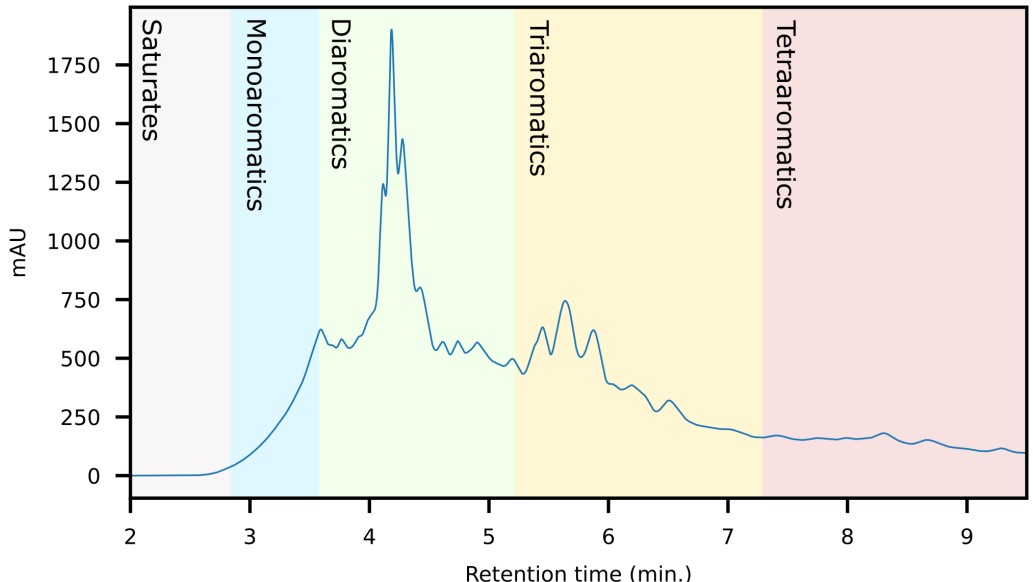

**Figure 5  HPLC-Chromatogram (with UV response measured at 272 nm) of a representative crude oil showing the approximate retention regions of the different aromatic classes.**

In terms of spectral quality, an approximately 10-fold reduction in background noise is observed in the LC fractions as compared to SPE. A comparison of the extracted mass spectra for the peak corresponding to 1,3-dimethylpyrene is presented in Fig. 7. Selected ion monitoring can be used to reduce background interferences for target species but results in loss of spectral detail for qualitative analysis. Furthermore, when using low-resolution instruments, *i.e.*, single quadrupole MS, there is a large risk of overlap in complex samples (*Rosenthal, 1982*; (*Davis & Giddings, 1983*). The reduction in background noise improved library matching, especially for analytes present at trace levels.

To evaluate identification performance, peak picking and library matching were carried out using MassHunter Unknowns Analysis and the NIST mass spectral library. The match factor limit was set to 700. The number of compounds was compared both on a sample-to-sample basis and by merging all features from all samples (with duplicates removal based on CAS number). A comparison of the merged compound tables of all samples shows that
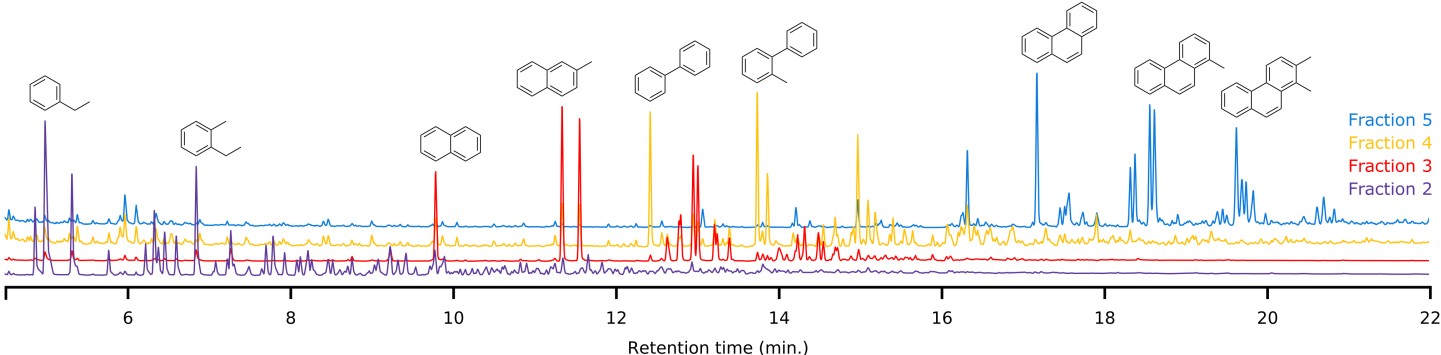

**Figure 6** **Offset overlay of the GC-MS chromatograms of fractions 2 to 5 obtained using the LC method of a representative sample.** Typical compounds in each fraction have been marked to demonstrate the separating power.

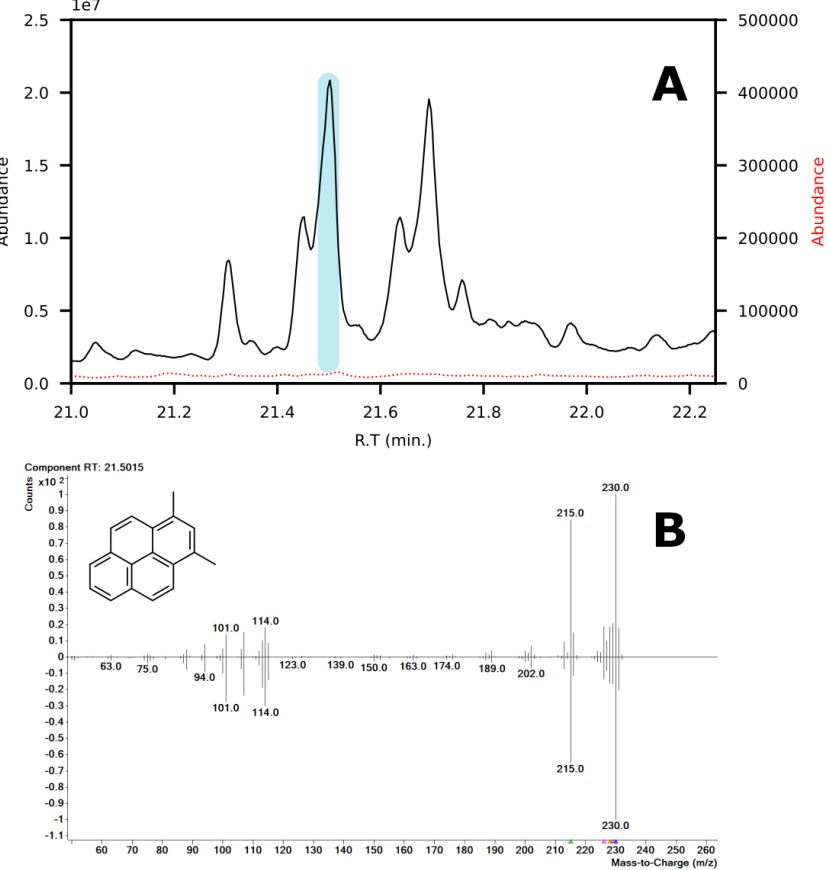

**Figure 7** **(A) Chromatograms showing the peak of 1,3-dimethylpyrene (marked in blue) found in LC-fraction 7 (black, pre-concentrated) and the aromatics fraction obtained via SPE (dotted red). (B) Mass spectra mirror plot of the LC-fraction peak showcasing its purity.**

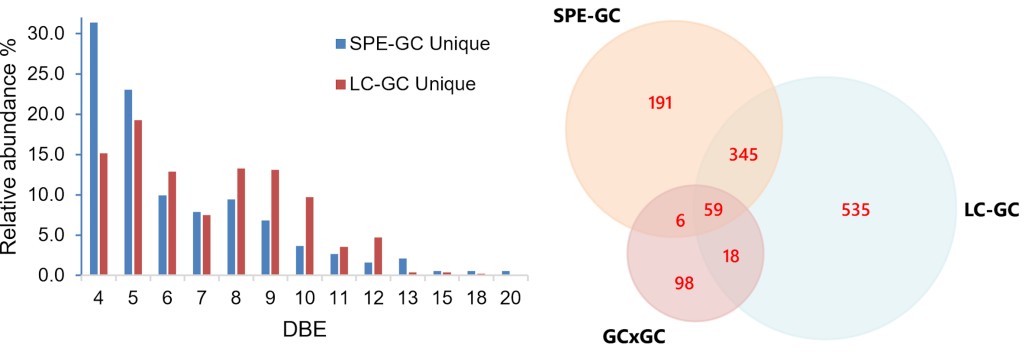

**Figure 8** **Double bond equivalent (DBE) distribution table for assigned unique compounds obtained from SPE-GC and LC-GC methods.** The Venn diagram shows the total amount of assigned compositions obtained from SPE-GC-, LC-GC- and GC×GC-MS.

using the LC-GC method we can identify 957 compared to 601 compounds using SPE. This is an increase of 37.2%. To increase the match confidence, we applied a retention index (RI) filter, only retaining compounds with a match within 100 units of the library value. By doing so, we identified 426 compared to 300 (42% increase). This excludes all compounds of which a library RI is not available (approximately 1% of our feature set). However, a large portion of the compounds only have computationally approximated retention indices and not experimentally determined values. Thus, all filtering and data analyses should be carried out with care and manual intervention. The SPE fraction contains approximately 190 unique compounds with 404 compounds overlapping both analyses. Manual inspection reveals this list contains several petroleum-type compounds and not predominantly background noise or contamination (*e.g.*, plasticizers, column contamination). One plausible source is errors occurring during the automatic processing routines. Small differences in mass spectra (*e.g.*, due to abundance or background level) can lead to closely related library matches being given similar (but different) priority (*e.g.*, isomeric species). Figure 8 shows the DBE distributions of assigned compounds uniquely observed from SPE-GC and LC-GC methods. Noticeably, the DBE distributions are significantly different between the two methods. The distribution of unique compounds from SPE-GC is centered around low DBE 4 and 5 (*e.g.*, monoaromatic), whereas unique compounds from LC-GC are distributed more evenly at higher DBE values. This is expected as the LC-GC method isolates and enriches high aromaticity fractions. These findings showcase the ability of LC-GC as a high-resolution fractionation method for crude oil.

For comparison to comprehensive multi-dimensional chromatography, the samples were analyzed by our in-house routine GC × GC-MS method (*i.e.*, solvent dilution, filtration and analysis) (Fig. 9). The objective of the GC × GC method is not to maximize feature ID but enable multi-class analysis/fingerprinting with minimal to no sample preparation. Furthermore, the SPE-LC-GC and GC × GC analyses were carried out on different instruments which makes direct comparison challenging. For GC × GC an Agilent 7200B QTOF high-resolution mass spectrometer was used. For SPE-LC-GC, an Agilent

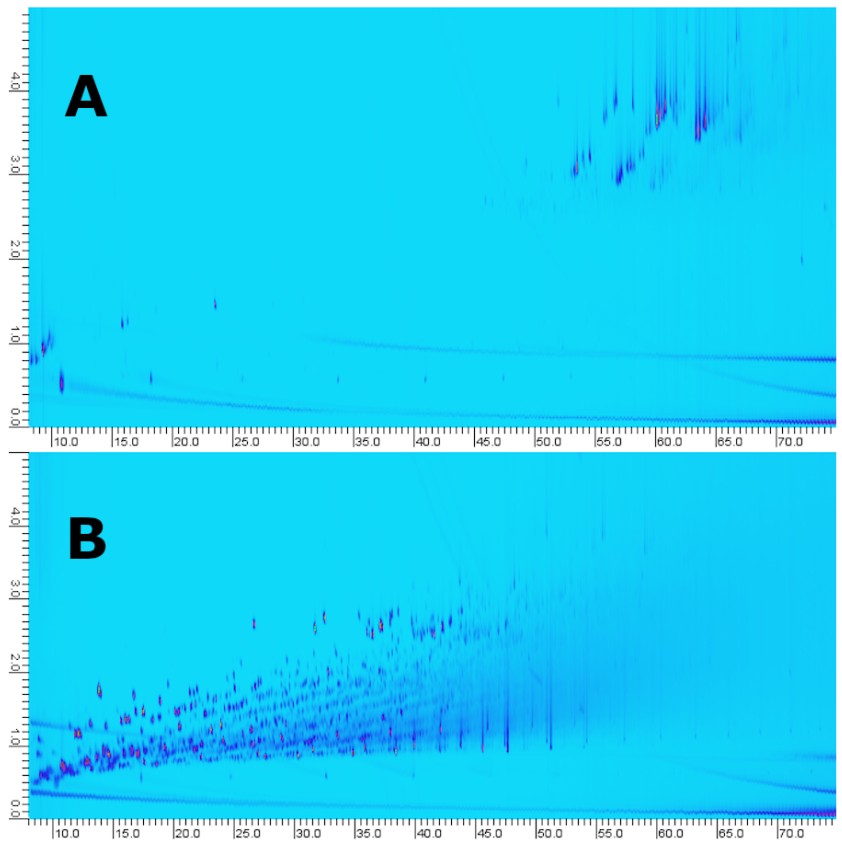

**Figure 9   GC×GC-Chromatograms of fraction 5 obtained using the LC-method (A) compared to whole oil analysis (B, *i.e.*, no pre-fractionation).** The LC-fraction has a high abundance of alkyl phenanthrene isomers at or below the LOD seen in whole oil GC×GC-MS.

5977B single quadrupole equipped with a High-Efficiency Source (HES) was used. The HES has both higher sensitivity and dynamic range. Secondly, for GC × GC, the dilution factor was adjusted so that the analytes with the highest abundance were at detector saturation. Here, we see that although GC×GC is not restricted in terms of peak capacity, it does fall short in terms of dynamic range. After blob detection, library matching and filtering we obtain 63 tentative hits in a single sample. With corresponding processing settings, we identified 143 compounds by multi-fraction LC-GC-MS analysis of the same sample. This is an increase of 127%. In terms of manual intervention, ease of use and time of analysis, GC × GC is preferred compared to LC-GC-MS. However, the amount of data generated using the latter is more comprehensive in our case.

A Venn diagram was constructed to compare three methods (Fig. 8). The number within each colored circle represents the number of assigned unique compounds for each method, whereas numbers in overlapped zones represent the number of compounds that have been co-assigned from corresponding methods. The amount of compositions obtained from LCxGC (957) significantly surpasses GC × GC (181) and SPE-GC (601). We obtained approximately 50% unique compounds with GC × GC and LC-GC and 32% with SPE-GC.

**Table 3 Comparison of the number of tentatively identified species in SPE-GC and GCGC versus LC-GC.** The calculations are based on merged data of all samples with the removal of duplicates. Unique indicates compounds not identified in the compared methods.

| Filter | GC×GC Tot. | GC×GC Unique | SPE-GC Tot. | SPE-GC Unique | LC-GC Tot. | LC-GC Unique |
|---|---|---|---|---|---|---|
| No formula or R.I filter | 181 | 98 | 601 | 191 | 957 | 535 |
| No formula filter, R.I $\pm$ 100 units | 82 | 15 | 300 | 68 | 426 | 187 |
| N,S,O $\geq$1, no R.I filter | 80 | 79 | 311 | 151 | 517 | 357 |
| N,S,O $\geq$1, R.I $\pm$ 100 units | 6 | 6 | 91 | 44 | 124 | 81 |

The Venn diagram also shows that co-assigned compounds of those three methods cover a narrow range of overall chemical composition (59 co-assigned compounds) of crude oil. Again, it is worth noting that there are differences in terms of dilution factor and instrumental parameters for those methods. Therefore, the comparison is biased but still relevant to evaluate the LC-GC method for trace components analysis.

## CONCLUSIONS

We have developed a method for high-resolution fractionation of complex crude oil matrices. By using sub-micron LC columns we obtained high efficiency and resolution which allowed intra-class compound separation. This is in contrast with traditional methods, *e.g.*, SPE, which yields a single aromatics fraction. The method is especially advantageous for the isolation of trace species. Multiple compounds not observed by SPE-GC-MS were pre-concentrated yielding high abundance and spectral quality. The increase in the number of tentatively identified peaks is thus a result of both reduced co-elution and an increase in analyte signal-to-noise ratio.

By characterization of 14 crude oils, we extended the identification to a large number of hydrocarbon and N,S,O-containing aromatics. Of the 517 uniquely identified compounds, 69% (357) contain either N,S,O (or a combination of) atoms (Table 3). The structures of five representative compounds are presented in Fig. 10. Aromatic nitrogen and sulfur compounds are detrimental in petroleum processing. Furthermore, they potentially have biological activity and may pose an environmental and toxicological hazard (*López García et al., 2002*; *Anyanwu & Semple, 2015*; *Zhang et al., 2018*; *Vetere, Pröfrock & Schrader, 2021*). Therefore, their characterization is an important pursuit. They are routinely analyzed by direct infusion mass spectrometry that provides the molecular formula but not connectivity (*Guan, Marshall & Scheppele, 1996*; *Purcell et al., 2007b*; *Purcell et al., 2007a*; *Corilo, Rowland & Rodgers, 2016*). Thus, isolation and GC-MS analysis with library matching provide valuable information on their presence in oil samples.

The relatively long fractionation time (60 min) and the number of fractions generated lead to a full sample analysis time of 6 h (when characterizing the first 7 fractions using GC-MS). Several steps require manual intervention, *i.e.*, dilution and pre-concentration of fractions and moving the samples from the LC to the GC. It would therefore be beneficial to implement more automation, *e.g.*, by using liquid handling robotics (ultimately with direct hyphenation to the GC). We observed minor co-elution during the analysis of

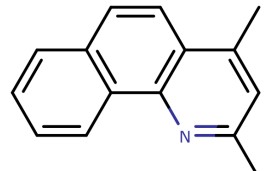

Indan-5-carboxylic acid
**Fraction 2**

1-Naphthylamine
**Fraction 3**

4'-Ethyl-4-biphenylcarboxylic acid
**Fraction 4**

2,4-Dimethylbenzo[h]quinoline
**Fraction 5**

8-Methyl-benzo[b]naphtho[2,3-d]thiophene
**Fraction 6**

**Figure 10** **The molecular structures of five unique compounds identified in separate fractions (2 to 6) obtained by using the LC-method.**

latter fractions. Combining the LC-fractionation with subsequent GC × GC analysis would increase the power of the method further. However, it would require an intense data processing workflow with high demands in computational power. Something that is already challenging in comprehensive GC × GC studies (*Reichenbach et al., 2019*; *Wilde et al., 2020*; *Stefanuto, Smolinska & Focant, 2021*).

# ACKNOWLEDGEMENTS

The authors are grateful for the donation of the samples from Total E&P Denmark. Furthermore, the authors wish to thank Karen L. Feilberg for discussions.

## Funding

This work was supported by the Danish Hydrocarbon Research and Technology Center. The funders had no role in study design, data collection and analysis, decision to publish, or preparation of the manuscript.

## Grant Disclosures

The following grant information was disclosed by the authors:
Danish Hydrocarbon Research and Technology Center.

## Competing Interests

The authors declare there are no competing interests.

## Author Contributions

- Khoa Huynh conceived and designed the experiments, performed the experiments, analyzed the data, prepared figures and/or tables, authored or reviewed drafts of the paper, and approved the final draft.
- Annette E. Jensen conceived and designed the experiments, performed the experiments, authored or reviewed drafts of the paper, and approved the final draft.
- Jonas Sundberg conceived and designed the experiments, performed the experiments, analyzed the data, performed the computation work, prepared figures and/or tables, authored or reviewed drafts of the paper, and approved the final draft.

## Data Availability

The raw data (GC-MS, feature tables) are available at Zenodo: Sundberg, Jonas. (2021). Extended characterization of petroleum aromatics by off-line LC-GC-MS (dataset) [Data set]. Zenodo. https://doi.org/10.5281/zenodo.5121065.

## Supplemental Information

Supplemental information for this article can be found online at http://dx.doi.org/10.7717/peerj-achem.12#supplemental-information.

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
