# Peer review of "Extended characterization of petroleum aromatics using off-line LC-GC-MS"

_PeerJ Analytical Chemistry, doi:10.7717/peerj-achem.12_

## Round 0.1 · original submission · Minor Revisions

Firstly, please accept my apologies for the delay with this paper. Unfortunately it proved very difficult to get reviewers.

The good news is that I now have two sets of comments and both recommend minor revisions.

·

Basic reporting

This is an interesting paper describing a new selective and efficient method for high-resolution fractionation of complex mixtures of petroleum aromatics which are contained in crude oils in a wide range of concentrations. The method is based on semi-preparative liquid chromatography and subsequent GC-MS analysis of collected fractions. The method enables not only an efficient isolation of the more abundant aromatics, but also the separation, isolation and preconcentration of trace aromatic components. This significantly increases the number of aromatic species which can be identified in crude oil samples compared to traditional methods like SPE-GC-MS or comprehensive GCxGC.

The manuscript is written in professional, unambiguous language. The article includes a sufficient introduction and the relevant prior literature is appropriately referenced. I consider the paper suitable for publication after some corrections in order to improve its quality. Specific remarks are listed below.

Experimental design

The subject is well within the scope of Peer J An. Chem and the manuscript structure conforms to PeerJ standards.

The method and all experiments are clearly described and the method is properly validated. The advantages and disadvantages (e.g. relatively long fractionation time) of the method are pointed out and well documented. The presented results confirm that the method can serve as a complementary tool to traditional sample preparation techniques or comprehensive GCxGC. HPLC fractionation of 14 crude oils with subsequent GC-MS analysis demonstrated the method suitability for identification of trace aromatics including the tentative identification of several compounds not observed by comparable routinely used methods.

Validity of the findings

All relevant data, on which the conclusions are based, are provided in tabulated (three tables) or graphical form (ten figures). The conclusions are safe and well supported by experimental results.

Additional comments

Specific remarks:

1. Experimental results are presented in three tables but two of them (Tables 1 and 2) are not mentioned anywhere in the text relevant for Table 1 (lines 216 to 230, and for Table 2 (lines 239 to 242).

2. There is a discrepancy between the figure numbers (Figures 5–9) listed in related manuscript paragraphs and the numbers and content of attached figures:
line 244: Figure 4 should be corrected to Figure 5
line 245: Figure 5 should be corrected to Figure 6
line 274: Figure 6 should be corrected to Figure 7
lines 301 and 328: Figure 7 should be corrected to Figure 8
line 312: Figure 8 should be corrected to Figure 9
Consequently, figure number should be corrected also in the Supplemental file:
Figure_5_Full_chromatograms to Figure_6_Full_chromatograms

3. In Table 3 title GCxGC should be added. The first sentence should read: Comparison of the number of tentatively identified species in SPE-GC and GCxGC versus LC-GC.

4. Figure 1: SPE enables separation of saturates and aromatics (not saturates and saturates).

5. Line 72: corect (M. Pampanin & O. Sydnes, 2017) to (Pampanin & Sydnes, 2017)

6. Lines 145 and 146: correct µM to µm

7. Line 236: correct BTEX-type monoaromatics (benzene, toluene, xylene) to BTEX-type monoaromatics (benzene, toluene, ethylbenzene, xylenes)

8. Line 244: delete the word secondary

9. Line 270: correct (Countryman, Kelly & Garriques) to (Countryman, Kelly & Garriques, 2005)

10. Lines314 and 316: abbreviation SPE/LC-GC is confusing; I would suggest writing SPE-GC and LC-GC instead.

11. References:
Line 408: correct Countryman S, Kelly K, Garriques M. to Countryman S, Kelly K, Garriques M. 2005.

12. Line 410: correct Davis JM, Giddings JCalvin. to Davis JM, Giddings JC.

13. Line 442: correct M. Pampanin D, O. Sydnes M to Pampanin DM, Sydnes MO.

Reviewer 2 ·

Basic reporting

Tables 1 and 2 are not referenced in the body of the text

Figure 9 is not referenced in the text

The supplemental figure 6 has a file name denoting figure 5

Line 181 - there is an erroneous "29,30"

Is the Python script you used for the data processing available?

The figure legends need more information. As they stand, the figures require far too much reader intepretation to guess what they are showing.

Experimental design

An interesting premise, a DB-5 column was used for the GC portion of the experiment. Why was this preferred to the more routinely used DB-1 column? Or even a PAH column as this separates all isomers? Would using a different column have an impact on the identification on isolated chemicals?

You used one type of SPE sorbent for the SPE experimentation. Were other SPE sorbents tested and did these have any affect on the extraction?

Validity of the findings

no comment

---

## Round 0.2 · accepted · Accept

Thank your revising your manuscript in response to the reviewers comments and concerns.

·

Basic reporting

The authors have successfully revised the original manuscript by taking into account all the reviewers' suggestions and comments. In the accompanying letter they clearly explained all the changes made in the manuscript text, tables, and figures. They have also addressed and satisfactorily commented some issues raised by the reviewers such as the choice of the GC column and SPE sorbent. In my opinion this manuscript can be accepted for publication in the present form.

Experimental design

No comment.

Validity of the findings

No comment.

Additional comments

No comment.